# Strategies for Offline Adaptive Biology-Guided Radiotherapy (BgRT) on a PET-Linac Platform

**DOI:** 10.3390/cancers17152470

**Published:** 2025-07-25

**Authors:** Bin Cai, Thomas I. Banks, Chenyang Shen, Rameshwar Prasad, Girish Bal, Mu-Han Lin, Andrew Godley, Arnold Pompos, Aurelie Garant, Kenneth Westover, Tu Dan, Steve Jiang, David Sher, Orhan K. Oz, Robert Timmerman, Shahed N. Badiyan

**Affiliations:** 1Department of Radiation Oncology, University of Texas Southwestern Medical Center, Dallas, TX 75390, USAtu.dan@utsouthwestern.edu (T.D.);; 2RefleXion Medical, Hayward, CA 94545, USA; 3Department of Radiology, University of Texas Southwestern Medical Center, Dallas, TX 75390, USA; orhan.oz@utsouthwestern.edu

**Keywords:** BgRT, PET-linac, PET signal loss, offline adaptive radiotherapy strategies

## Abstract

This study is the first to define and evaluate offline adaptive radiotherapy (ART) workflows specifically for biology-guided radiotherapy (BgRT) on a PET-linac system. Three offline adaptation strategies were proposed, each tailored to distinct clinical scenarios. Two clinical cases were presented to demonstrate the feasibility of two of these strategies. As more clinical centers adopt BgRT, the proposed workflows will help manage treatment variability and mitigate risks associated with adaptive planning.

## 1. Introduction

Incorporating biological information into radiotherapy has long been a highly active area of research and clinical development in the field of radiation oncology [1,2,3]. Historically, most efforts have focused on utilizing PET/CT as the primary modality to provide biological or functional insights, predominantly in offline treatment settings. Offline PET/CT images have been widely used for initial target delineation and for replanning to account for anatomical changes or treatment response [2,4,5,6,7].

More recently, a novel radiotherapy platform—RefleXion X1 (RefleXion Medical, Hayward, CA, USA)—has been introduced, integrating a PET/CT scanner with a medical linear accelerator (linac) [8,9]. This system enables a new treatment paradigm known as Biology-guided Radiotherapy (BgRT), commercially branded as SCINTIX^®^ therapy. The X1 system with BgRT from RefleXion leverages real-time PET signals emitted by the tumor to dynamically guide the delivery of radiation beamlets. Tumor-generated PET signals act as biological fiducials, allowing for highly precise radiation targeting and offering a novel solution for motion management. BgRT eliminates the need for large motion-encompassing margins used in internal target volume (ITV)-based approaches, potentially allowing for significant target volume reduction, margin tightening, and dose escalation, thereby improving outcomes and reducing treatment toxicity [3,8]. Because PET signals reflect underlying tumor molecular activity, BgRT also holds the potential for functional adaptation during treatment. A recent Investigational Device Exemption (IDE) clinical trial (BIOGUIDE-X NCT04788147) involving FDG with lung and bone lesions demonstrated the safety and early clinical efficacy of this approach, leading to the FDA clearance of BgRT/SCINTIX therapy for treating tumors in these anatomical sites [10].

To ensure the safe delivery of BgRT plans, the RefleXion X1 system incorporates several built-in safety mechanisms. A key feature is a pre-treatment PET acquisition (“pre-scan”) performed before each fraction. The pre-scan evaluates tumor PET uptake, PET signal quality, and predicts dose distribution to determine whether the original BgRT plan can be safely delivered. Treatment proceeds only if all predefined criteria are satisfied. However, during the course of several treatment fractions, tumor metabolic and structural changes, such as tumor shrinkage or growth, may alter PET signals, necessitating replanning to maintain treatment safety and efficacy. As of now, the RefleXion X1 platform does not offer direct online adaptation but supports an offline adaptive workflow.

This paper introduces practical strategies for offline adaptation on the RefleXion X1 PET-linac, focusing on two clinical scenarios:(a)Tumor structural changes without significant changes in PET signal, where normal tissues remain stable;(b)Changes in PET signal (with or without accompanying structural modifications), again with stable normal tissue anatomy.

In the *Materials and Methods* section, we introduce the BgRT workflow and outline practical strategies for offline adaptation. We then present two clinical cases to demonstrate the feasibility of this approach in the *Results* section, followed by a discussion of its limitations and future directions. To our knowledge, this paper is the first clinical evaluation of adaptive BgRT using the PET-Linac system.

## 2. Materials and Methods

### 2.1. Introduction to the RefleXion X1 System

The RefleXion X1 system is a novel radiotherapy platform integrating a 6 MV flattening filter-free (FFF) linac with real-time PET imaging [9]. Two PET detectors are mounted orthogonally to the beamline on an O-ring gantry which rotates at 60 revolutions per minute (RPM). The PET detectors are 5.1 cm in the SI direction and capable of reconstructing 3D PET images every 100 milliseconds, enabling continuous monitoring of the patient’s PET biodistribution [11]. The on-board kVCT system operates in clinical mode using a helical acquisition with a fixed tube potential of 120 kVp while allowing for adjustable mAs and pitch settings. Multiple clinical protocols are available and can be selected based on the treatment site [12]. The linac delivers a nominal dose rate of 1000 MU/min at an 85 cm source-to-axis distance (SAD). Beam shaping is achieved using a split-jaw system (10 mm or 20 mm widths) and a binary multileaf collimator (MLC) consisting of 64 leaves, each 6.25 mm wide, capable of leaf switching in under 10 milliseconds. Patient positioning is managed along six degrees of freedom using the couch and gantry rotation (five-degree-of-freedom (5DoF) for the couch). The couch advances by 2.1 mm increments during treatment delivery (axial mode).

PET data acquisition begins by converting the electronic signals from the SiPM into singles events, encoding the hit location, energy, and timing. Coincidence events are identified from these singles for image reconstruction. Random coincidences are estimated using a delayed window approach and corrected by subtracting Gaussian-smoothed random data in sinogram space. Due to real-time delivery constraints, no explicit scatter correction is applied; however, scatter is partially suppressed using an energy window of 395–600 keV. For image reconstruction and quality evaluation, filtered back-projection (FBP) with a Hanning filter is used following single-slice re-binning, leveraging the system’s short axial field of view (5.3 cm) to enable efficient 2D reconstruction. Data are corrected for attenuation, decay, and randoms, but not scatter. Attenuation correction is based on the 511 keV PET attenuation map derived from the simulation CT. The same correction and reconstruction protocols are applied for both imaging and treatment modes.

The system supports two clinical modalities: IGRT, Image-Guided Radiotherapy without PET-based guidance, and BgRT (SCINTIX^®^ therapy), Biology-Guided Radiotherapy utilizing real-time PET signals. SCINTIX therapy is FDA-cleared for FDG-guided treatment of lung and bone lesions.

### 2.2. BgRT Planning and Delivery Workflow

Figure 1 illustrates a typical workflow for BgRT planning and delivery using the RefleXion X1 system [10]. Similar to conventional radiotherapy, the process begins with a CT simulation (CT sim). For targets subject to motion, a 4D CT is required to capture the full range of motion.

During the contouring phase, in addition to delineating the target and organs at risk (OARs), a unique structure called the Biological Tracking Zone (BTZ) must be defined. The BTZ is a conceptual volume that encompasses the entire range of target motion with added margins. It plays a critical role in BgRT, as the system will restrict beam delivery to PET-avid regions within the BTZ while masking out all PET activity outside this zone. The definition and optimization of the Biology-Tracking Zone (BTZ) are beyond the scope of this study. Further details on BTZ construction and evaluation can be found in the literature [8,9,10].

An additional step in the BgRT planning workflow is the PET functional modeling (FM) session. During an FM session, the patient is injected with the prescribed dose of PET tracer (15 mCi for FDG). Following tracer administration, the patient is set up on the RefleXion X1 system, and kVCT imaging is used to align the patient with the planning CT. PET images are then acquired during this imaging-only session (no radiation is delivered at this stage). BgRT planning can commence only after the FM session is complete.

The final dose distribution is driven by both standard dosimetric constraints and PET data from the FM session. To assess the reliability and stability of the PET signal, two key metrics **Activity Concentration (AC)** and **Normalized Target Signal (NTS)**, are calculated and provided directly to users in RefleXion TPS.

**Activity Concentration (AC)**: This indicates the absolute signal strength available to guide beam delivery. AC is calculated as the difference between the mean value in the top 20% of the PET voxels in the BTZ and the mean value of voxels in a 3 mm shell outside the BTZ.**Normalized Target Signal (NTS)**: This measures the contrast of the tumor signal relative to the immediate background noise. NTS is the ratio of the AC to the standard deviation of the voxels in the 3 mm shell outside the BTZ.

Planning may proceed only if the following thresholds are met: AC > 5 kBq/ml and NTS > 2.7.

Once a BgRT plan is generated, it undergoes both clinical and physics review. A unique feature of the RefleXion treatment planning system (TPS) is the bounded dose-volume histogram (bDVH), which is computed for the BTZ and OARs. This concept, analogous to a robustness evaluation in proton therapy [13], simulates a variety of plausible uncertainties during BgRT delivery, including

Variations in FDG uptake of ±25% relative to the planning PET;Rigid setup shifts within the BTZ up to 5 mm in any direction;Dose uncertainty of 3%.

The bDVH provides clinicians with both visual and quantitative tools to assess best-case and worst-case dosimetric scenarios, thus improving confidence in treatment robustness. The final plan then follows standard approval and quality assurance (QA) procedures before clinical delivery.

For each BgRT treatment fraction, a short pre-scan PET is acquired following kVCT alignment but before treatment delivery. This scan is used to reassess the PET signal’s stability for that session. The AC and NTS metrics are recalculated, and a predicted dose distribution is generated by applying the original plan’s beam data to the pre-scan data. This predicted dose distribution is compared to the original planned dose distribution to evaluate consistency. To quantify the agreement between the predicted and planned dose distributions, the **bDVH%** passing rate is used. This metric calculates the percentage of DVH points for the OARs and BTZ that fall within the bounds of the originally approved bDVH.

**bDVH%**: This quantifies the DVH point agreement dose for OARs and BTZ between bDVH from the initial plan and from the predicted dose on treatment day based on pre-scan PET.

Treatment may proceed only if the following thresholds are met: AC > 5 kBq/ml, NTS > 2.0, and bDVH passing rate > 95%. Again, RefleXion TPS calculates and provides these three parameters to the user. These thresholds were determined based on internal validation, including imaging studies and robustness requirements of the tracking and dose-delivery algorithms. Among the three, AC and NTS are evaluations based directly on PET signal quality, while bDVH% is derived from dose prediction metrics. In general, failures in AC and NTS are more challenging to recover from, as they often indicate more significant signal degradation.

In addition, three more parameters, **Target Contrast at FM (TC_FM)**, **Target Contrast Ratio during Treatment (TCR_Tr)**, and **mean activity concentration in the BTZ (MeanBTZ)**, can be manually calculated in an offline setting using any imaging processing tools to further evaluate and compare the relative PET uptake in the tumor:**Target Contrast at Functional Modeling (TC_FM)**: This denotes the contrast between the PET count in the PTV with respect to the PET counts outside the PTV but within the BTZ. TC_FM is computed as the relative difference between the mean activity concentration in the PTV and mean PET activity within the BTZ (excluding the PTV), divided by the mean PET activity within the PTV.**Target Contrast Ratio during Treatment (TCR_Tr)**: This checks if the PET contrast in PTV with respect to the background within the BTZ (excluding the PTV) on the day of treatment is similar to that observed during functional modeling. TCR_Tr is calculated as the ratio of the target contrast observed during pre-scan to that during FM (planning).**Mean PET count in the BTZ (MeanBTZ)**: This ensures that there is sufficient PET uptake within the BTZ for planning and treatment. MeanBTZ represents the mean activity concentration in the BTZ. If the MeanBTZ value is higher than 100 kBq/ml, then we cap it at 100 kBq/ml, as it means there are more than sufficient PET counts for accurate treatment.

These metrics are valuable for further evaluating and understanding signal changes. Trends and variations in these parameters can help predict when the system will trigger an interlock that prevents the system from proceeding with the BgRT plan.

### 2.3. Strategies for Offline Adaptive Workflow

Figure 2 illustrates the potential outcomes of the pre-treatment assessment—including localization CT and pre-scan PET—for BgRT fraction N, along with the corresponding offline adaptation strategies.

***Scenario 0***: Minor or no changes in anatomical structure and/or PET uptake values—treatment continues.***Scenario 1***: Reduction in tumor size—Offline Adaptation Strategy 1, preemptive adaptation, is useful for structural changes in the shape of the tumor, while the shape of other organs in the patient remains the same. The change in tumor size could cause a variation in the pre-scan PET uptake values compared to FM PET.***Scenario 2***: Large PET Parameter Variation—Offline Adaptation Strategy 2, partial re-simulation, can be used when PET parameters such as AC or NTS changes moderately approaching or at threshold, or when bDVH approaches 95%.***Scenario 3***: Significant changes in patient anatomy and PET signal—Offline Adaptation Strategy 3, full re-simulation, can be used when there is a significant change in the overall anatomical shape of the patient and/or a very significant change in the PET uptake parameters.***Scenario 4***: BgRT treatment halted due to significant loss of PET signal—When AC and/or NTS drop well below their respective thresholds, BgRT treatment cannot proceed. In this situation, a separately generated IGRT plan without PET guidance may be implemented to continue the patient’s treatment course.

**Figure 2 cancers-17-02470-f002:**
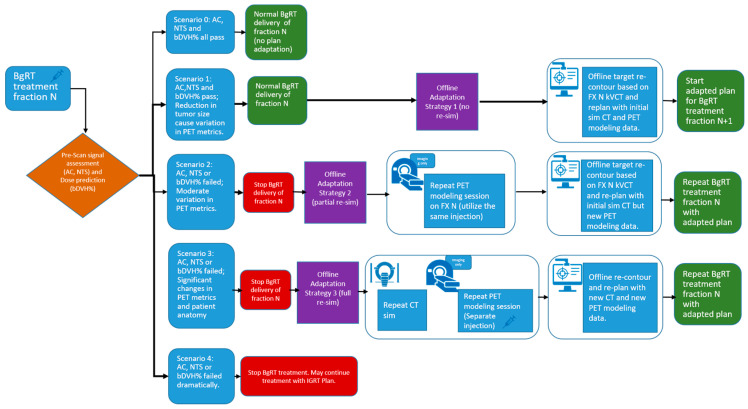
Decision pathways after pre-scan PET assessment at fraction N. Five scenarios are presented. Three offline adaptive strategies and their workflow to address scenarios 1–3 are presented. Syringe icon indicates injection is needed.

As mentioned previously, early treatment responses or changes in patient anatomy can affect FDG uptake (reflected in AC and NTS) or cause deviations in dose distribution (bDVH%). If the observed changes are minor and all thresholds are met, treatment with the original BgRT plan can continue as described in Scenario 0. However, if one or more criteria are not satisfied, the original plan cannot be delivered, and an offline adaptive workflow may be required.

In general, three offline strategies are proposed when patient anatomy changes and/or failed AC, NTS, or bDVH% values are observed:***Offline Adaptation Strategy 1***: Preemptive adaptation (reduction in tumor size, small variation in PET metrics, plan still deliverable).

This is described in Scenario 1. If all parameters pass, the original BgRT plan for fraction N can proceed. However, if the size of the tumor reduces on localization CT while the size and shape of all the other organs remains the same, then Scenario 1 can be used. The change in tumor size and PET uptake values may indicate an increased likelihood of failure in upcoming fractions. In such cases, a preemptive adaptation strategy can be implemented without a re-simulation. Using the kVCT images acquired during fraction N, contours can be revised—particularly if structural changes are observed affecting the target or OARs—and the plan can be re-optimized using the original CT simulation dataset, as well as original PET data from the FM session. This approach is suitable when minor anatomical changes (e.g., tumor shrinkage or growth) are present but FDG uptake remains well above threshold. Density overrides may be applied to reflect structural changes. The newly adapted plan can then be used for fraction N+1.

***Offline Adaptation Strategy 2***: Partial re-simulation (failed metrics, moderate signal or structural changes).

As shown in Scenario 2, if AC, NTS, TCR_Tr, MeanBTZ, or bDVH% drops significantly for fraction N, the original BgRT plan cannot be delivered. In this case, a partial re-simulation strategy may be appropriate if the patient’s overall anatomical changes are minor. A new PET FM session on the RefleXion X1 can begin immediately after the failed pre-scan using the same radiotracer injection, thus avoiding the need for an additional injection if rescheduled efficiently. Updated contours can be drawn based on the kVCT from fraction N, and the newly acquired PET data are used for re-optimization while still referencing the original CT simulation dataset. This strategy is suitable when metabolic signal changes (e.g., from tumor response) cause PET parameter failure but FDG uptake itself is still robust and above delivery threshold. The adapted plan can then be used to repeat fraction N, offering the benefit of avoiding an extra patient visit and minimizing additional tracer exposure.

***Offline Adaptation Strategy 3***: Full re-simulation (failed metrics, significant signal and structural changes).

In this case (Scenario 3), the patient’s anatomy has changed significantly, necessitating a full re-simulation. This scenario can happen when the patient loses a lot of weight and the tumor changes significantly with respect to the initial simulation CT. This scenario may or may not be associated with a significant change in the PET uptake values. This process essentially restarts the planning workflow, including a new CT simulation, a new PET FM session, and a fully re-optimized plan. A new FM PET is needed as the simulation CT is used to generate the attenuation map used for the PET reconstruction during FM and for treatment. The patient will require an additional tracer injection for the new PET FM session.

Offline Adaptation Strategy 3 essentially mirrors the standard BgRT workflow. Therefore, in this study, we focus on Strategies 1 and 2. In the *Results* section, we demonstrate the feasibility of offline adaptation for Strategies 1 and 2 through real clinical cases, highlighting how they support the continuity of BgRT treatment while maintaining safety and efficacy. All patients included in this study received appropriate regulatory approvals, including institutional review board clearance (IRB, study number STU-2021-0976)

## 3. Results

### 3.1. Offline Adaptation Strategy 1: Preemptive Adaptation

A tumor in the left upper lung (LUL) was treated with BgRT for a total dose of 50 Gy delivered in five fractions. Tumor shrinkage was observed at fraction 4. During the pre-scan of the fourth fraction, the activity concentration (AC) measured 44.51 kBq/mL, the normalized target signal (NTS) was 24.81, and the bDVH passing rate was 100%. The PET evaluation metrics of AC, NTS, and bDVH passed the required thresholds, allowing the BgRT treatment to proceed as planned. However, when compared to the planning PET data, a 36% decrease in AC and a 31% decrease in NTS were noted.

During the offline analysis, tumor contour (GTV) was re-drawn on kVCT, which revealed a 55% volume reduction compared to the original contour (6.8 cc at Fraction 4 vs. 12.8 cc from the initial plan). Although PET metrics had AC > 5 kBq/ml and NTS > 2, the trend, as well as the reduction in tumor volume in fraction 4, suggested a potential risk of failure in subsequent fractions. Therefore, a preemptive adaptive plan was generated as a precautionary measure for fraction 5. Table 1 summarizes the AC, NTS, TC_FM, TCR_Tr, and MeanBTZ for the patient over the course of the treatment. This patient is a good representative case in which AC, NTS, and TCR_Tr show a good response to treatment, with each fraction showing the tumor is not radioresistant. In this case, all fractions were successfully completed. However, a gradual decrease in PET uptake was observed over the initial three fractions. A preemptive adaptation was performed with fraction 4 data to account for the reduction in tumor size observed in the localization CT. The overall weight and shape of the organs in the patient’s body was relatively the same for all fractions. Hence, Scenario 1 of the adaptive BgRT plan was used for this patient.

Figure 3 illustrates the original CT simulation and PET FM images alongside the kVCT and PET images acquired during fraction 4. A new gross tumor volume (GTV) was delineated on the original CT dataset, using the kVCT images from fraction 4 as guidance. Additionally, a density override structure (set to lung-equivalent density) was applied around the tumor to account for changes due to tumor shrinkage and to maintain an accurate dose calculation in the re-optimized plan.

The newly adapted planning target volume (PTV) measured 22 cc, compared to 39.5 cc in the original plan, representing nearly a 50% reduction in volume. The re-planning process followed the same requirements and instructions as the initial plan: the prescription dose was required to cover at least 95% of the PTV, while all OARs met institutional dose constraints.

Figure 4 presents a comparison between the initial and adapted plans. The adapted plan demonstrates a smaller irradiation volume with tighter isodose lines, resulting in a reduction in OAR doses. Because the same PET FM data were used, the overall shape of the isodose distribution remained similar to that of the original plan. However, the adapted plan also reset the PET uptake values based on the updated anatomy and contour changes.

### 3.2. Offline Adaptation Strategy 2: Partial Re-Simulation

A left upper lung (LUL) tumor was treated with BgRT to a total dose of 60 Gy in five fractions. Significant variation in PET uptake, as well as anatomical shrinkage of the tumor, was observed at fraction 3. Signal uptake assessment was failed during PET signal evaluation based on pre-scan PET. Therefore, the initial BgRT plan could not be delivered. In response, the treatment team quickly assessed the situation during fraction 3 and elected to initiate an immediate PET FM session, using the same FDG injection and enabling a seamless transition into an offline adaptive workflow. The patient successfully completed the new PET FM session without requiring an additional 18F-FDG PET tracer injection.

During the offline analysis process, it was revealed that, comparing the localization CT with the SimCT, the tumor had anatomically shrunk from 19.2 cc to 8.3 cc, resulting in a ~57% reduction in volume. Activity concentration (AC) measured 11.92 kBq/mL, the normalized target signal (NTS) was 5.06; MeanBTZ = 5.89 kBq/ml and TCR_Tr = 0.724. Compared to the planning PET data (AC = 23.61 kBq/mL, NTS = 15.68, and MeanBTZ = 37.87 kBq/mL), this represented a substantial drop of 50% in AC, 68% in NTS, 27.6% in TCR_Tr, and 84% in MeanBTZ, indicating a significant change in tumor biology. Table 2 shows the PET evaluation parameters over the course of the treatment. A partial adaptation was performed and new structures for GTV, PTV, and BTZ were drawn. A small increase in AC, NTS, and MeanBTZ was observed in the new FM, most likely due to the revised volume.

As described previously, when there is a significant variation in the PET signal, the system interlocks and stops treatment. The AC, NTS, and bDVH values are not presented to the user when this happens. At this time, the user can quickly acquire a new FM PET using the same injected dose. After the new FM session, the pre-scan PET image, localization CT, SimCT, FM PET, and contours are loaded in an image processing tool that enables manual checking of the PET image. Further, new GTV, PTV, BTZ, and OARs can be drawn if needed, and the new PET uptake values can be recalculated and compared to those obtained using the original FM PET.

As shown in Table 2, it is a good idea to check whether the bDVH is dropping with each fraction. This is often a tell-tale sign that the tumor is responding well to radiotherapy and the PET uptake in the tumor cells might warrant a new FM to account for the lower PET uptake in future fractions. Another step the user can take is to check whether the PET signal in the tumor is very low, to the point that contrast between the mean PET uptake in the tumor with respect to the background is less that 2:1. For these cases, the TCR_Tr and MeanBTZ should also result in very low values, representing a successful response to BgRT treatment. A low uptake of PET in the tumor after a few BgRT fractions could be due to the fact that most of the PET-avid tumor cells are dead and are responding well to the treatment.

Figure 5 shows a comparison of images from the original and adapted BgRT workflows. A new gross tumor volume (GTV) was delineated on the original CT dataset using the kVCT acquired during fraction 3 for anatomical guidance. The new GTV measured 8.3 cc, compared to 19.12 cc in the original plan, reflecting a significant reduction in tumor volume. A density override structure was also created to account for the volumetric change and was assigned a lung-equivalent density to ensure accurate dose calculation. A new treatment plan was then generated using the updated CT, as well as updated PET data from the FM session, allowing the treatment to continue with improved accuracy and alignment to the tumor’s current state.

Figure 6 presents a dosimetric comparison between the initial and adapted plans for this case. In the original plan, the target coverage was compromised to spare nearby critical structures, specifically the brachial plexus and the ribs, as the tumor was close to both OARs. The adapted plan was re-optimized to maintain equivalent PTV coverage while respecting institutional dose constraints for these OARs. The final dose distribution in the adapted plan reveals a smaller irradiated volume and a sharper dose fall-off, resulting in lower OAR doses within the defined volumetric constraints. Because new PET data were acquired to guide re-optimization, the shape of the dose cloud differed from that of the original plan, reflecting the updated metabolic landscape of the tumor.

## 4. Discussion

In this study, we introduce a clinical workflow for biology-guided radiotherapy (BgRT) using the RefleXion PET-linac platform. Recognizing that tumor structural and metabolic changes may occur during the treatment course, we describe the safety mechanisms designed to ensure the robustness and deliverability of BgRT. These mechanisms include assessment of AC, NTS, and the bDVH passing rate during plan creation and immediately prior to treatment. When there is a change to the tumor size, patient anatomy, and/or PET uptake, then plan adaptation may be necessary to maintain safe and effective treatment delivery. The new BgRT plan will be created only if AC > 5 kBq/mL and NTS > 2.7, while treatment is possible only if AC > 5 kBq/mL, NTS > 2.0, and bDVH passing rate > 95%.

To address these scenarios, we propose and discuss three offline adaptive strategies, each tailored to different clinical circumstances. To our knowledge, this is the first work to define and evaluate offline adaptive radiotherapy (ART) workflows specifically for BgRT. With more clinical centers adopting BgRT treatment, we hope these strategies will aid in managing treatment variations and mitigating risks related to adaptive planning.

Each of the proposed three strategies has its own advantages and trade-offs, which we summarize below.

Offline Adaptation Strategy 1 (preemptive adaptation) is seamless for the patient and can be implemented in anticipation of expected changes in upcoming fractions. Instead, it leverages the localization kVCT acquired during treatment setup and reuses the original CT simulation data. A trend analysis of AC, NTS, TCR_Tr, MeanBTZ, and bDVH% can help determine when preemptive adaptation is warranted. This strategy is best suited for cases involving minor tumor changes with stable body geometry and when PET signal metrics remain well above threshold values.

Regarding Offline Adaptation Strategy 2 (partial re-simulation), if any of the key PET assessment parameters reduces drastically and AC and NTS are still above the PET evaluation thresholds, then a partial re-simulation can be performed. A new PET FM session can be initiated immediately using the same injected dose of FDG, saving the patient an additional visit and avoiding unnecessary radiation exposure. However, quick decision-making is essential, as delays post-injection can lead to a reduction in PET signal. Fortunately, the short pre-treatment PET scan used in BgRT makes same-day PET modeling feasible. This strategy is ideal for cases where tumor-related PET signal or structural changes are observed, but the rest of the patient’s anatomy remains stable.

In both Strategies 1 and 2, density override structures may be used to improve the accuracy of dose optimization. The Results section demonstrates the feasibility of these two strategies, including workflow implementation, updated contouring, and dosimetric comparison. 

When multiple assessment parameters fail and there are major anatomical changes or a significant loss of PET signal, in spite of a passing PET evaluation (AC, NTS, bDVH), a full re-simulation, Offline Adaptation Strategy 3, may be required. This approach mirrors the process for new patient treatment planning and includes a new CT simulation, new PET FM session, and plan generation. As this workflow is already detailed in the general BgRT method section, we did not include a clinical case example for this scenario.

Both Scenario 2 and 3 are triggered by failed pre-treatment evaluations (e.g., AC, NTS, or DVH%). The distinction lies in the extent of anatomical change. Scenario 2 applies when anatomy is largely stable—such as isolated tumor shrinkage—where a repeat functional modeling (FM) session may suffice. Scenario 3 is used when significant anatomical changes occur (e.g., weight loss, OAR deformation), requiring re-simulation and FM to reset the PET signal baseline. While Strategy 3 is more comprehensive, Strategy 2 is a practical, patient-friendly option when anatomical integrity is preserved. It helps reduce the number of clinic visits and minimizes additional FDG injections.

Several limitations of the current offline adaptive workflow should be acknowledged. Because there is typically at least a one-day gap between offline plan generation and the next treatment fraction, anatomical or metabolic changes could occur in the interim, potentially rendering the adapted plan suboptimal. This highlights the need for true online adaptation capabilities, which are not yet supported by the RefleXion platform. The overall concept and general workflow of online adaptation—including imaging, re-delineation, re-planning, and quality assurance—should follow standardized guidelines and recommended best practices [14,15].

We acknowledge that presenting only two cases to demonstrate the proposed strategies is a limitation of this study. However, our goal is to introduce a practical and robust framework for addressing the clinical challenges associated with PET signal loss in SCINTIX/BgRT treatments. Despite the limited number of cases, the scenarios, strategies, and workflows described are intended to be broadly applicable to similar clinical situations. As SCINTIX/BgRT technology continues to be adopted in practice, we hope this study will initiate and contribute to ongoing discussions among early adopters on the effective use of adaptive approaches to manage PET signal loss.

Additionally, both offline Strategies 1 and 2 rely on the original CT simulation data, which may introduce dose calculation inaccuracies if anatomical changes are significant. Clinical teams should remain mindful of this uncertainty during plan evaluation. Furthermore, in the proposed workflow, localization kVCT is used only to assist in target and OAR delineation. While, in theory, the RefleXion kVCT images could be used directly for planning, this functionality is not currently supported.

Of course, in cases where dramatic changes in tumor uptake are observed—such as complete loss of PET signal—none of the proposed offline adaptive strategies would likely be effective or sufficient to restore treatment feasibility. In these situations, as highlighted in Scenario 4, an IGRT plan without PET guidance can be generated to continue the treatment course.

While the primary focus of this manuscript is to propose strategies for addressing changes in PET signal and restoring deliverability, anatomical changes between fractions also have a significant role in triggering offline adaptation. In such cases, a repeat CT scan and a new session for functional modeling are typically required, aligning more closely with a traditional IGRT-based adaptive workflow. As mentioned before, online adaptation, particularly using kVCT guidance available on this platform once available, may offer a more efficient and responsive approach to managing anatomical variations in real time.

## 5. Conclusions

Offline adaptation plays a critical role in ensuring the safety, robustness, and continuity of Biology-Guided Radiotherapy (BgRT) treatments on the RefleXion X1 platform, particularly in the absence of online adaptation capabilities. In this study, we propose and evaluate three clinically actionable offline adaptive strategies—**preemptive adaptation, partial re-simulation, and full re-simulation**—to address dynamic changes in tumor biology and patient anatomy over the treatment course.

Each strategy offers a unique balance of practicality, resource use, and clinical effectiveness:

**Preemptive adaptation** enables proactive intervention without the need for additional imaging or tracer injection.

**Partial re-simulation** allows for same-day functional modeling signal acquisition using the same FDG injection, providing a patient-friendly and resource-efficient option.

**Full re-simulation** mirrors traditional re-planning workflows and is reserved for cases with major anatomical and metabolic changes.

To our knowledge, this is the first study to define and categorize offline adaptive workflows specifically for PET-guided BgRT, incorporating PET-specific evaluation metrics (AC, NTS, bDVH%) into clinical decision-making. While the limited number of cases precludes statistical analysis, our findings demonstrate the technical feasibility and potential dosimetric advantages of these strategies, particularly when PET signal variability threatens treatment deliverability.

Although offline adaptation introduces inherent limitations—such as delayed implementation and reliance on prior imaging—these strategies represent an important step toward systematic and standardized adaptive workflows for BgRT. As future developments enable true online PET-guided adaptation, these offline approaches provide a strong foundational framework for managing intra-treatment variability in a clinically feasible and safe manner.

## Figures and Tables

**Figure 1 cancers-17-02470-f001:**
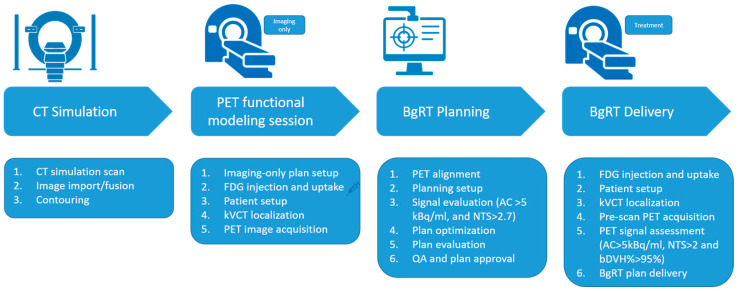
BgRT planning and deliver workflow. Four major steps: CT sim, PET functional modeling (FM) session, BgRT planning, and BgRT plan delivery. The sub-steps are also described for each major steps.

**Figure 3 cancers-17-02470-f003:**
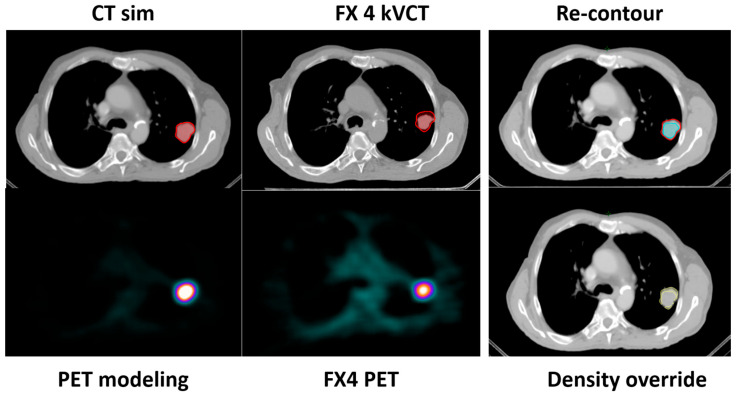
Images and contours used for BgRT planning and re-planning. Compared to SimCT, the tumor volume (GTV) shrunk by 55%. Original (red) and new (blue) GTV contours are highlighted on sim CT. A density override structure (yellow) is also shown to compensate for the tumor shrinkage.

**Figure 4 cancers-17-02470-f004:**
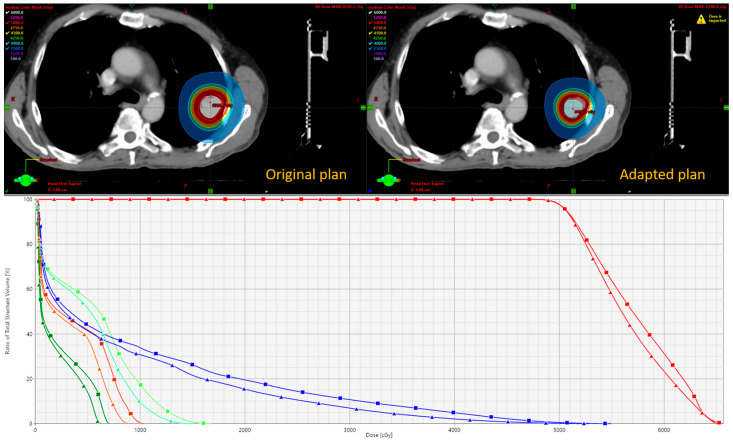
Dosimetric comparison of original and adapted plan. Top panel shows dose distribution with representative isodose lines. Bottom panel shows DVHs of trachea (orange), spinal cord (dark green), ribs (blue), aorta (light green), and PTVs (red). Square, original plan. Triangle, adapted plan.

**Figure 5 cancers-17-02470-f005:**
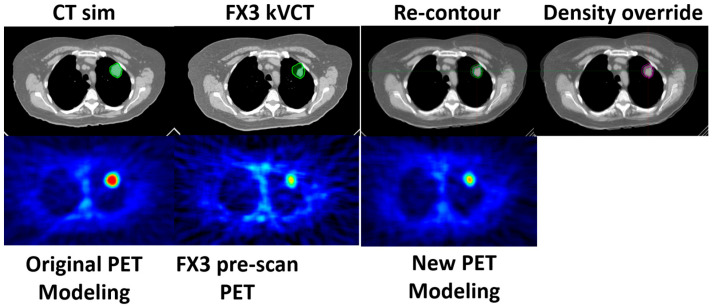
Visualization of tumor changes in Planning SimCT, localization kVCT, and PET images, as well as contours used for BgRT planning and re-planning. A significant difference in target volume was observed in the localization kVCT of FX3, along with a degraded pre-scan PET signal. Original (light green) and new (dark green) GTV contours are highlighted on the sim CT. A density override structure (purple) is also shown to compensate for the tumor shrinkage. The new PET FM session image is also presented for comparison.

**Figure 6 cancers-17-02470-f006:**
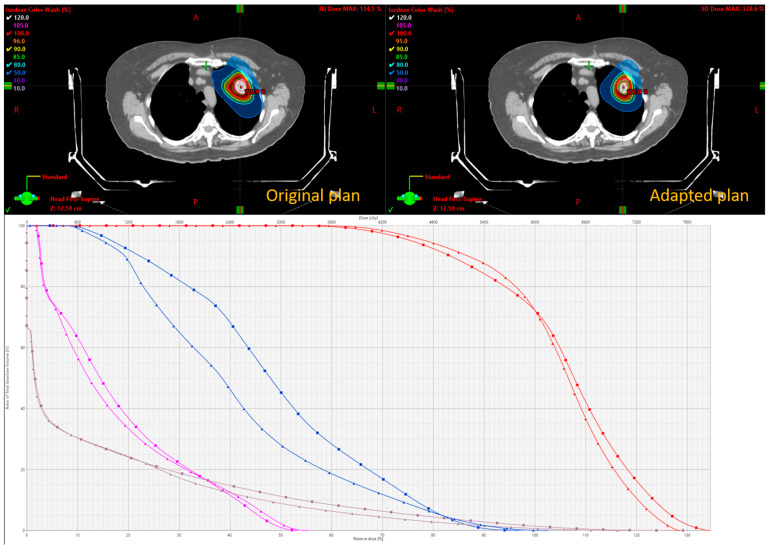
Dosimetric comparison of original and adapted plan. Top panel shows dose distribution with representative isodose lines. Bottom panel shows DVHs of brachial plexus (pink), ribs (blue), left lung (brown), and PTV (red). Square, original plan. Triangle, adapted plan.

**Table 1 cancers-17-02470-t001:** PET uptake values of a tumor in the left upper lung. AC and NTS values reduced by fraction 3 and were mostly stable, while MeanBTZ reduced significantly by the fifth fraction. On the other hand, the size of the tumor reduced significantly in fraction 4, making this an ideal candidate for Scenario 1 of adaptive BgRT delivery.

Type	Fraction	Days Post FM	AC (kBq/mL)	NTS	bDVH (%)	TC_FM	TCR_Tr	MeanBTZ (kBq/mL)
FM plan		0	61.30	36.23	-	1.16	1.00	>100
Treatment	1	9	60.99	30.43	100	-	0.99	>100
Treatment	2	13	48.74	27.59	100	-	1.01	>100
Treatment	3	15	39.88	24.68	100	-	0.94	>100
Treatment	4	20	44.51	24.81	100	-	0.95	>100
Treatment	5	23	33.49	18.13	100	-	0.88	65.91

**Table 2 cancers-17-02470-t002:** PET uptake values for a tumor in the left upper lung. AC, NTS, TCR_Tr, and MeanBTZ reduced significantly for the third fraction, making this an ideal candidate for Scenario 2 of the adaptive BgRT plan.

	Fraction	Days Post FM	AC (kBq/mL)	NTS	bDVH (%)	TC_FM	TCR_Tr	MeanBTZ (kBq/mL)
FM, Plan		0	23.61	15.68	-	1.01	1.00	37.87
Treatment	1	8	23.60	14.06	98.5	-	0.97	28.72
Treatment	2	12	20.52	8.78	96.1	-	0.89	10.47
Treatment	3	15	11.92	5.06		-	0.72	5.90
New FM, Adapted plan		0	14.26	8.95	-	0.92	1.00	8.95

## Data Availability

The original contributions presented in this study are included in the article. Further inquiries can be directed to the corresponding author(s).

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
