# Peer review of "Strategies for Offline Adaptive Biology-Guided Radiotherapy (BgRT) on a PET-Linac Platform"

_cancers, 2025, doi:10.3390/cancers17152470_

Round 1
Reviewer 1 Report
Comments and Suggestions for Authors
This manuscript introduces a structured workflow for offline adaptive Biology-Guided Radiotherapy (BgRT) using a PET-linac system. The topic is relevant, and the authors present their methods clearly, supported by clinical case examples that demonstrate feasibility. The technical description of the system and decision-making process is comprehensive and well illustrated.
The manuscript is overall well written, though it would benefit from minor language editing to improve clarity. The conflict of interest section is transparent, though the authors might consider explicitly clarifying any affiliations with the device manufacturer. Additionally, while limitations are mentioned, a brief comment on the potential impact of anatomical changes between fractions could further strengthen the discussion.
Author Response
Reviewer 1:
Comment1: This manuscript introduces a structured workflow for offline adaptive Biology-Guided Radiotherapy (BgRT) using a PET-linac system. The topic is relevant, and the authors present their methods clearly, supported by clinical case examples that demonstrate feasibility. The technical description of the system and decision-making process is comprehensive and well illustrated.
Thanks for the comment.
Comment2: The manuscript is overall well written, though it would benefit from minor language editing to improve clarity.
Thanks for the comment. We went through the manuscript focusing on the language and made revision to improve the clarity.
Comment3: The conflict of interest section is transparent, though the authors might consider explicitly clarifying any affiliations with the device manufacturer.
Thanks for the comment. We revised the COI accordingly.
Comment4: Additionally, while limitations are mentioned, a brief comment on the potential impact of anatomical changes between fractions could further strengthen the discussion.
We appreciate the reviewer’s insightful comment. While the primary focus of this manuscript is to propose strategies for addressing changes in PET signal and restoring deliverability, we agree that anatomical changes between fractions also play a significant role in triggering offline adaptation. In such cases, a repeat CT scan and a new session for functional modeling are typically required, aligning more closely with a traditional IGRT-based adaptive workflow. As noted in the discussion, online adaptation—particularly using kVCT guidance available on this platform—may offer a more efficient and responsive approach to managing anatomical variations in real time.
We added the following to the manuscript in the discussion section:
“ While the primary focus of this manuscript is to propose strategies for addressing changes in PET signal and restoring deliverability, anatomical changes between fractions are also a significant role in triggering offline adaptation. In such cases, a repeat CT scan and a new session for functional modeling are typically required, aligning more closely with a traditional IGRT-based adaptive workflow. As mentioned before, online adaptation, particularly using kVCT guidance available on this platform once available, may offer a more efficient and responsive approach to managing anatomical variations in real time.”
Reviewer 2 Report
Comments and Suggestions for Authors
Dear Authors,
This study proposes offline adaptive radiotherapy strategies for BgRT applications based on PET-Linac system. Three different adaptation strategies guided by parameters such as AC, NTS and bDVH% were defined on RefleXion X1 device and exemplified with two clinical cases. The following corrections should be taken into consideration.
-There is a limited number of cases. The applicability of the strategies is presented only on two clinical cases. This situation limits the generalizability of the results.
-The strategy selection process is not explained. The decision algorithm regarding which strategy should be selected at which thresholds is not clear. In particular, the distinction between "Scenario 2" and "Scenario 3" can be blurred in practice.
-Although the limitations of offline methods are accepted, a comparative evaluation with online BgRT systems has not been made. This does not clarify the advantage-disadvantage balance of the proposed strategies.
-The equivalents of metrics such as AC, NTS, bDVH% in the clinical decision-making process are poorly explained. In particular, it is not clear whether the threshold values ​​are empirical or clinical.
-Although the concept of Biological Tracking Zone is important, it is not clear how its clinical equivalent is determined and how its boundaries are optimized.
-Although the tables and figures are sufficiently explanatory, some are insufficiently referenced in the text. In particular, the titles of Table 1 and Table 2 are missing or irregular.
-Although dosimetric comparisons are shown in the figure, no statistical analysis is performed. It is not clearly stated whether the adaptation provides a significant dose advantage.
This study presents one of the first sets of strategies for clinical application for offline adaptation of PET-linac-based BgRT and fills an important gap. However, it is open to criticism due to the limited number of cases, potential for conflict of interest, insufficient justification of threshold values, and lack of comparison of strategies with online methods. It needs to be supported by more comprehensive clinical data and technology-independent validation studies. The article will be improved if the above-mentioned deficiencies are addressed.
Best regards
Author Response
Reviewer 2:
Comment 1: There is a limited number of cases. The applicability of the strategies is presented only on two clinical cases. This situation limits the generalizability of the results.
Thank you for the comment. We agree with the reviewer that the limited number of clinical cases is a constraint of this study. However, our intent is to present a practical and robust framework to address the clinical challenge of PET signal loss in SCINTIX/BgRT treatments. While only two cases are presented, the scenarios, proposed strategies, and associated workflows are designed to be broadly applicable to similar situations encountered in clinical practice.
As the field gradually adopts SCINTIX/BgRT technology, we anticipate a learning curve in managing treatment disruptions due to signal variability. We hope this manuscript will contribute to that evolving discussion among early adopters. In addition to the proposed adaptive strategies, we believe our explanation and discussion of multiple metrics used to evaluate PET signal quality provide valuable insights into machine behavior and the underlying algorithms that support BgRT functionality.
To address this concern, we added the following content to the manuscript in discussion session.
“We acknowledge that presenting only two cases to demonstrate the proposed strategies is a limitation of this study. However, our goal is to introduce a practical and robust framework for addressing the clinical challenges associated with PET signal loss in SCINTIX/BgRT treatments. Despite the limited number of cases, the scenarios, strategies, and workflows described are intended to be broadly applicable to similar clinical situations. As SCINTIX/BgRT technology continues to be adopted in practice, we hope this study will initiate and contribute to ongoing discussions among early adopters on the effective use of adaptive approaches to manage PET signal loss.”
Comment 2: The strategy selection process is not explained. The decision algorithm regarding which strategy should be selected at which thresholds is not clear. In particular, the distinction between "Scenario 2" and "Scenario 3" can be blurred in practice.
Thank you for the comment. We agree that the distinction between Scenario 2 and Scenario 3 can be nuanced in clinical practice, and we appreciate the opportunity to clarify the strategy selection process. Both scenarios share the common trigger of failing pre-treatment evaluations, such as AC, NTS, or DVH% checks. The key distinction lies in the underlying cause of the PET signal discrepancy. Scenario 2 applies when no significant anatomical changes are observed—for example, when the tumor has decreased in size but the overall patient anatomy remains stable. In these cases, a repeat functional modeling (FM) session alone may be sufficient to restore the PET signal baseline. Scenario 3, on the other hand, is appropriate when substantial anatomical changes are present, such as patient weight loss, deformation of nearby OARs, or changes in skin contour. These cases require a full re-simulation with a new CT scan and a subsequent FM session to re-establish the reference baseline.
While Strategy 3 is a more comprehensive approach and could theoretically be used in all situations, Strategy 2 offers a more practical and patient-friendly solution when anatomical stability is confirmed. It helps reduce the number of clinic visits and minimizes additional FDG injections. We have clarified this distinction and the decision-making algorithm in the revised manuscript.
We added the following paragraph in discussion session to address this concern
“Both scenario 2 and 3 are triggered by failed pre-treatment evaluations (e.g., AC, NTS, or DVH%). The distinction lies in the extent of anatomical change. Scenario 2 applies when anatomy is largely stable—such as isolated tumor shrinkage—where a repeat functional modeling (FM) session may suffice. Scenario 3 is used when significant anatomical changes occur (e.g., weight loss, OAR deformation), requiring re-simulation and FM to reset the PET signal baseline. While Strategy 3 is more comprehensive, Strategy 2 is a practical, patient-friendly option when anatomical integrity is preserved. It helps reduce the number of clinic visits and minimizes additional FDG injections.”
Comment 3: Although the limitations of offline methods are accepted, a comparative evaluation with online BgRT systems has not been made. This does not clarify the advantage-disadvantage balance of the proposed strategies.
Thank you for the comment. As noted in the manuscript, online BgRT adaptation solutions are currently not available on the PET-linac platform used in this study. Therefore, a direct, side-by-side comparison between offline and online adaptive strategies is not feasible at this time. We have discussed the limitations of the proposed offline methods in the manuscript. Looking ahead, we plan to collaborate with vendors to explore and evaluate online adaptive solutions as part of our future work.
Comment 4: The equivalents of metrics such as AC, NTS, bDVH% in the clinical decision-making process are poorly explained. In particular, it is not clear whether the threshold values ​​are empirical or clinical.
We appreciate the reviewer’s comment. AC, NTS, and bDVH% are PET signal evaluation metrics developed by the vendor, with threshold values hard-coded into the system software. These thresholds were determined based on internal validation, including imaging studies and robustness requirements of the tracking and dose-delivery algorithms. Among the three, AC and NTS are evaluations based directly on PET signal quality, while bDVH% is derived from dose prediction metrics. In general, failures in AC and NTS are more challenging to recover from, as they often indicate more significant signal degradation.
We add the following paragraph in Material and Method session
“These thresholds were determined based on internal validation, including imaging studies and robustness requirements of the tracking and dose-delivery algorithms. Among the three, AC and NTS are evaluations based directly on PET signal quality, while bDVH% is derived from dose prediction metrics. In general, failures in AC and NTS are more challenging to recover from, as they often indicate more significant signal degradation.”
Comment 5: Although the concept of Biological Tracking Zone is important, it is not clear how its clinical equivalent is determined and how its boundaries are optimized.
We appreciate the reviewer’s insightful comment. The Biology-Tracking Zone (BTZ) represents the full extent of target motion with an additional margin to account for biology-guidance uncertainty and patient setup error. Based on benchtop testing, a 5 mm total margin has been found to be sufficient. Similar to the Internal Target Volume (ITV), the BTZ is defined using a 4D-CT image set acquired during simulation, capturing tumor positions throughout the respiratory cycle. However, unlike the ITV, the BTZ is not a treatment volume. Instead, it defines the spatial boundary from which PET emissions are collected to guide radiotherapy delivery. This helps restrict guidance to the target region and minimizes the influence of PET-avid non-target structures.
We acknowledge that the clinical determination and optimization of BTZ boundaries are important topics, but a detailed discussion is beyond the scope of the current study. Relevant information can be found in prior literature (e.g., Ref. 8, Ref. 9, Ref. 10).
To address this comment, we have added the following clarification to the Methods and Materials section:
“The definition and optimization of the Biology-Tracking Zone (BTZ) are beyond the scope of this study. Further details on BTZ construction and evaluation can be found in the literature [Ref8-10].”
Comment 6: Although the tables and figures are sufficiently explanatory, some are insufficiently referenced in the text. In particular, the titles of Table 1 and Table 2 are missing or irregular.
Thanks for the comment. We went through, revised the manuscript, and confirmed all figures and tables are appreciated referenced.
Comment 7: Although dosimetric comparisons are shown in the figure, no statistical analysis is performed. It is not clearly stated whether the adaptation provides a significant dose advantage..
Thank you for the comment. The primary objective of this study is to propose offline adaptation strategies aimed at restoring plan deliverability following the failure of AC, NTS, or DVH% criteria. Achieving a statistically significant dosimetric advantage was not the primary focus or an expected outcome of the adapted plans. That said, the two clinical cases presented did demonstrate dosimetric improvements as a secondary benefit, owing to updated contours, revised PET signal, and plan re-optimization.
Given the limited sample size of two cases, we did not perform statistical analysis, as it would not yield meaningful conclusions regarding the dosimetric advantage of adaptation.
We have clarified this point in the revised manuscript in the discussion session.
“The primary objective of this study is to propose offline adaptation strategies aimed at restoring plan deliverability following the failure of AC, NTS, or DVH% criteria. Achieving a statistically significant dosimetric advantage was not the primary focus or an expected outcome of the adapted plans. That said, the two clinical cases presented did demonstrate dosimetric improvements as a secondary benefit, owing to updated contours, revised PET signal, and plan re-optimization.”
Comment 8: This study presents one of the first sets of strategies for clinical application for offline adaptation of PET-linac-based BgRT and fills an important gap. However, it is open to criticism due to the limited number of cases, potential for conflict of interest, insufficient justification of threshold values, and lack of comparison of strategies with online methods. It needs to be supported by more comprehensive clinical data and technology-independent validation studies. The article will be improved if the above-mentioned deficiencies are addressed.
We appreciate the reviewer’s insightful comment. We acknowledge the limitations of this study and have addressed the reviewer’s concerns in the revised manuscript, which we believe has significantly improved the clarity and completeness of our work. We would also like to emphasize that proposing offline adaptation strategies is only the first step in addressing the clinical challenges associated with PET signal loss. As mentioned in the discussion, we hope our work will initiate further efforts and stimulate meaningful dialogue within the community. Moving forward, we plan to collaborate closely with the vendor to propose, test, and validate robust and clinically meaningful offline and online adaptation strategies, with the ultimate goal of maximizing the therapeutic benefit of this novel treatment for our patients.
Reviewer 3 Report
Comments and Suggestions for Authors
The authors presented the work titled as "Strategies for Offline Adaptive Biology-Guided Radiotherapy (BgRT) on a PET-Linac Platform". The overall quality and presentation of the manuscript sounds standard.
There are some concerns which authors should be addressed:
- There are some inconsistency in writing style which must be professionally rewritten such as in line number 184 ... "Figure 2 outlines possible outcomes from the pre-treatment (localization CT ... "
- It would be more interesting if the authors could added pseudocode or proper workflow for the entire study and also highlight in method section about some technical handling such as in Figure 3 FX4 PET.
- In discussion section, I will suggest to remove the subheadings.
- Conclusion appears to be too short and too summarized. There must be some specific conclusion and terms of finding or uniqueness of the method.
- There are several places where typos mistakes are present.
- The keywords does not cover the entire story of this article.
Author Response
Reviewer 3:
Comment 1: There are some inconsistency in writing style which must be professionally rewritten such as in line number 184 ... "Figure 2 outlines possible outcomes from the pre-treatment (localization CT ... )"
Thanks for the comment. We revised the line 184 as the following “Figure 2 illustrates the potential outcomes of the pre-treatment assessment—including localization CT and pre-scan PET—for BgRT fraction N, along with corresponding offline adaptation strategies.”
Comment 2: It would be more interesting if the authors could added pseudocode or proper workflow for the entire study and also highlight in method section about some technical handling such as in Figure 3 FX4 PET.
We appreciate the reviewer’s comment. We devoted considerable effort to developing a clear and effective way to present the three clinical scenarios and corresponding offline adaptation strategies. While we did consider using a pseudocode-style decision algorithm, we found that the complexity and verbosity required would make it difficult to convey concisely. Instead, we opted for a visual workflow, and we believe Figure 2 offers the most intuitive and comprehensive representation of the decision-making process and strategy selection. Although not in pseudocode form, this figure effectively captures the logic and structure of the proposed adaptive framework.
We add some technical details regarding PET imaging and construction in method section as followings
“PET data acquisition begins by converting the electronic signals from the SiPM into singles events, encoding the hit location, energy and timing. Coincidence events are identified from these singles for image reconstruction. Random coincidences are estimated using a delayed window approach and corrected by subtracting Gaussian-smoothed random data in sinogram space. Due to real-time delivery constraints, no explicit scatter correction is applied; however, scatter is partially suppressed using an energy window of 395–600 keV. For image reconstruction and quality evaluation, filtered back-projection (FBP) with a Hanning filter is used following single-slice re-binning, leveraging the system’s short axial field of view (5.3 cm) to enable efficient 2D reconstruction. Data are corrected for attenuation, decay, and randoms, but not scatter. Attenuation correction is based on the 511 keV PET attenuation map derived from the simulation CT. The same correction and reconstruction protocols are applied for both imaging and treatment modes.”
Comment 3: In discussion section, I will suggest to remove the subheadings.
Thanks for the comment. Subheadings are now removed.
Comment 4: Conclusion appears to be too short and too summarized. There must be some specific conclusion and terms of finding or uniqueness of the method.
We appreciate the reviewer’s insightful comment. Advantages and limitations are summarized and explained in details in discussion session. We re-write and expand the conclusions section to add more details as followings
“Offline adaptation plays a critical role in ensuring the safety, robustness, and conti-nuity of Biology-Guided Radiotherapy (BgRT) treatments on the RefleXion X1 platform, particularly in the absence of online adaptation capabilities. In this study, we propose and evaluate three clinically actionable offline adaptive strategies—preemptive adaptation, partial re-simulation, and full re-simulation—to address dynamic changes in tumor bi-ology and patient anatomy over the treatment course.
Each strategy offers a unique balance of practicality, resource use, and clinical effec-tiveness:
Preemptive adaptation enables proactive intervention without the need for addition-al imaging or tracer injection.
Partial re-simulation allows same-day functional modeling signal acquisition using the same FDG injection, providing a patient-friendly and resource-efficient option.
Full re-simulation mirrors traditional re-planning workflows and is reserved for cas-es with major anatomical and metabolic changes.
To our knowledge, this is the first study to define and categorize offline adaptive workflows specifically for PET-guided BgRT, incorporating PET-specific evaluation met-rics (AC, NTS, bDVH%) into clinical decision-making. While the limited number of cases precludes statistical analysis, our findings demonstrate the technical feasibility and po-tential dosimetric advantages of these strategies, particularly when PET signal variability threatens treatment deliverability.
Although offline adaptation introduces inherent limitations—such as delayed im-plementation and reliance on prior imaging—these strategies represent an important step toward systematic and standardized adaptive workflows for BgRT. As future develop-ments enable true online PET-guided adaptation, these offline approaches provide a strong foundational framework for managing intra-treatment variability in a clinically feasible and safe manner.”
Comment 5: There are several places where typos mistakes are present.
Thanks for the comment. We read through and revised the manuscript to fix typos.
Comment 6: The keywords does not cover the entire story of this article.
Thanks for the comment. We revised and add a couple of more keywords as the followings.
“BgRT; PET-linac; PET signal loss; Offline Adaptive Radiotherapy Strategies; “